# Effects of Thyme Essential Oil Microcapsules on the Antioxidant and Quality Characteristics of Mutton Patties

**DOI:** 10.3390/foods12203758

**Published:** 2023-10-12

**Authors:** Hongyan Yu, Zehao Ma, Jingyun Wang, Shiling Lu, Doudou Cao, Jiaxing Wu

**Affiliations:** 1School of Food Science and Technology, Shihezi University, Shihezi 832000, China; 13677558109@163.com (H.Y.); 13097516197@163.com (Z.M.); lushiling_76@163.com (S.L.); 13289939036@163.com (D.C.); wujiaxing031114@163.com (J.W.); 2Xinjiang Cerim Modern Agriculture Co., Shuanghe 833400, China

**Keywords:** thyme essential oil, microcapsules, phenolic compounds, mutton patties

## Abstract

This study aimed to assess the stability of thyme essential oil microcapsules (TEOMs) and their impacts on the antioxidant properties and quality of lamb patties. The results demonstrated that gum Arabic effectively enhanced the stability of phenols within the thyme essential oil (TEO), with an optimal core/wall ratio of 1:8. Substituting TEO with TEOMs in lamb patties led to reductions in the thiobarbituric acid content, carbonyl content, sulfhydryl loss, and protein cross-linking. Additionally, the TEOMs positively influenced the mutton patties’ color, texture, microbiological stability, and sensory attributes. These findings substantiate the idea that TEOMs exhibit significant potential as a natural preservative to enhance the quality of mutton patties.

## 1. Introduction

In recent years, the global market for ready-to-eat food has witnessed rapid growth due to the fast pace of modern society. Notably, mutton, which is a staple in the human diet, is renowned for its high protein content, low cholesterol levels, and well-balanced amino acid profile [1]. Among China’s most beloved meat products are mutton patties, which are known for their distinctive flavor and rich brown-red hue [2]. However, the processing conditions involved in preparing mutton patties, including heating and cold storage, can lead to the oxidation and degradation of meat components, resulting in a significant decline in quality. Research indicates that lipid and protein oxidation are the primary culprits behind the poor quality of mutton pies, leading to reduced nutritional value and the accumulation of harmful substances, such as aldehydes, ketones, and hydrocarbons [3]. Therefore, the inclusion of dietary antioxidants is an effective approach to mitigating the oxidation of mutton patties. While synthetic antioxidants are widely employed in the food industry, consumer concerns regarding their toxicity and carcinogenic potential have spurred interest in natural plant-derived antioxidants [4].

Thyme essential oil (TEO), which is derived from thyme (*Thymus mongolicus Ronn*), is commonly used as a culinary spice in Europe due to its distinctive aroma. TEO boasts an abundance of phenolic compounds, primarily carvacrol, thymol, p-cymene, and other valuable bioactive substances [5]. Research demonstrated that TEO possesses a range of pharmacological activities, including antibacterial, anti-inflammatory, antioxidant, flavor-enhancing, and heat-clearing effects [6]. Thymol, as the main component of thyme essential oil, has excellent antimicrobial effects against a wide range of microorganisms, such as Listeria monocytogenes, Escherichia coli, and Staphylococcus aureus [7]. Thyme essential oil is often used as a good antioxidant for food preservation. Ghafarifarsani et al. [8] found that improving the antioxidant capacity of fish by adding thyme essential oil to fish feed improved the health of fish. Zhang et al. [9] found that a new blend of thyme essential oil (curdlan/polyvinyl alcohol film) extends the shelf life of frozen meat by 10 days. Despite the well-established properties of TEO, its application in the food industry is limited by challenges such as low water solubility, poor thermal stability, pronounced volatility, and a pungent odor [6].

Microencapsulation has found widespread utility in the food-processing industry and biopharmaceuticals, offering benefits such as enhanced bioavailability and antibacterial efficacy, improved solubility and dispersibility, odor mitigation, and sustained release [10]. In response to the aforementioned limitations of TEO, microencapsulation has emerged as a promising alternative method. Recent studies showed that thyme essential oil microcapsules (TEOMs) can produce positive ameliorative effects in an in vitro mimetic system. For instance, Cai et al. [11] reported that microcapsule technology enhances TEO’s stability and water solubility. Additionally, Zhang et al. [6] found that TEOMs exhibit superior antibacterial properties compared with TEO. However, limited attention has been given to the antioxidant effects of TEOMs in mutton patties. In this study, TEO was encapsulated with gum Arabic, and the resulting TEOMs were applied to mutton patties during the heat treatment and storage phases. This research investigated the impact of TEOMs on various aspects, including the thiobarbituric acid content, carbonyl content, sulfhydryl levels, cross-linking, color, texture, microbiological stability, and sensory attributes of mutton pies.

## 2. Materials and Methods

### 2.1. Materials

Thyme essential oil was purchased from the Cedar Natural Vegetable Oil Co., Ltd., Jiangxi Province, China (the total content of phenolic in the TEO was 412.38 ± 0.14 mg gallic acid equivalent (GAE)/g). Gum Arabic was purchased from Chenya Biotechnology Co., Ltd., Henan, China. Fresh mutton was purchased from the local supermarket (Shihezi, China). All other chemicals and reagents were purchased from Solebo Technology Co., Ltd. (Beijing, China) and Sigma Chemicals (Steinheim, Germany).

### 2.2. Preparation of Thyme Essential Oil Microcapsules

The method of Devi, Das, and Badwaik was used with some modifications [12]. Gum Arabic (30 g) was completely dissolved in distilled water (90 mL) and stirred for 20 min with a magnetic stirrer (at 40 °C, 250 rpm) to prepare a gum Arabic emulsion. The core (TEO) and wall material (gum Arabic) were mixed at different ratios (core material:wall material was 1:2, 1:4, 1:6, 1:8, 1:10, and 1:12; *w*/*w*). The mixture was stirred and agitated with a magnetic stirrer for 2 h (at 40 °C, 250 rpm). It was then placed in a refrigerator for 24 h (at 4 °C) and vacuum filtered. The filter residue underwent five wash cycles with absolute ethanol to remove the essential oil adhering to the gum Arabic surface. Finally, the resulting precipitate was freeze-dried under vacuum conditions until a constant weight was achieved, producing TEOMs.

#### 2.2.1. Determination of Oil Content in Microcapsules

The dried microcapsules (0.1 g) were soaked in absolute ethanol and extracted with an ultrasonic extractor for 30 min (at 30 °C), and then the mixture was filtered. After diluting the filtrate to 10 mL with absolute ethanol, the absorbance was measured at 287 nm after multiple dilutions. The total oil content was obtained according to the standard curve regression equation. The embedding efficiency of the TEO was calculated using the following formula:(1)$$\mathrm{Embedding}\;\mathrm{efficiency}\;\mathrm{of}\;\mathrm{TEO}\;(\%)=\frac{\mathrm{Total}\;\mathrm{oil}\;\mathrm{content}\;\mathrm{in}\;\mathrm{microcapsules}}{\mathrm{Total}\;\mathrm{oil}\;\mathrm{content}\;\mathrm{in}\;\mathrm{emulsion}}\times 100\%$$

#### 2.2.2. Determination of Total Phenol Content

The total phenol content (TPC) was measured using the Folin–Ciocalteu method. The result is expressed in mg GAE/g. The embedding efficiency of total phenol was calculated according to the following formula:(2)$$\mathrm{Embedding}\;\mathrm{efficiency}\;\mathrm{of}\;\mathrm{TPC}\;(\%)=\frac{\mathrm{TPC}\;\mathrm{in}\;\mathrm{microcapsules}}{\mathrm{TPC}\;\mathrm{in}\;\mathrm{emulsion}}\times 100\%$$

### 2.3. Determination of the Stability of Phenols in TEOMs and TEO under Simulated Mutton Patty Processing Conditions

TEO and TEOMs were placed in glass culture dishes, heat-treated (initial oven temperature 160 °C for 15 min), and then stored in a refrigerator at 4 °C for 15 days (sampling on days 3, 6, 9, 12, and 15). The phenol content in the microcapsules was determined, and the loss rate of phenols was calculated according to the following formula:(3)$$\mathrm{Loss}\;\mathrm{of}\;\mathrm{TPC}\;(\%)=\frac{1\;-\;\mathrm{Final}\;\mathrm{phenol}\;\mathrm{content}}{\mathrm{Initial}\;\mathrm{phenol}\;\mathrm{content}}\times 100\%$$

### 2.4. Preparation of Mutton Patties

The fatty and lean meat was cut into small pieces and then ground into minced meat. The recipe for the lamb pie was mutton 850 g, fat 150 g, and salt 20 g. The mixture was stirred for 5 min with a stirrer (B7Lb, Shanghai Limai Machinery Co., Ltd., Shanghai, China). After mixing, the following four groups of mutton patties were prepared: control group (without any antioxidants), 2.5 g TEO (TEO treatment group), 1.25 g TEO and 10.84 g TEOMs (TEO + TEOMs treatment group), and 21.68 g TEOMs (TEOMs treatment group). Notably, the different additions of TEO and TEOMs were used to maintain an equivalent amount of phenolic compounds (1.03 g GAE/kg) in each sample, as was optimized in our previous research [13]. For each formulation, the mixture was stirred for an additional 20 min. Subsequently, the mixture was manually shaped into circular patties, each measuring 5 cm in diameter and 1 cm in thickness.

Then, the mutton patties were placed in the oven and baked for 15 min at an initial temperature of 160 °C. Once cooled to room temperature, the lamb patties were placed in polypropylene trays wrapped in polyvinyl chloride film (Jinhua Jinxiu Plastics Co., Ltd., Jinhua, China) and stored in a refrigerator for 15 days at 4 °C. Samples were taken at each sampling point (before the heat treatment; heat treatment; and storage for 3 days, 6 days, 9 days, 12 days, and 15 days) for further analysis.

### 2.5. Extraction of Phenolic Compounds from Mutton Patties

Phenolic compounds were extracted from the mutton patties, as described by Xu et al. [14]. Briefly, minced mutton patties (4 g) were added to deionized water (50 mL) and simulated gastric juice (50 mL, pH = 1.2, 0.4 M HCl with 8 g/L NaCI). The mixture was homogenized for 40 s (at 10,000 rpm), pig pepsin (10 mL, 4 g/L, pH 1.2) was added, and digestion took place in a water bath for 4 h (37 °C). Then, the mixture was filtered with four layers of medical gauze, and the supernatant was evaporated (at 50 °C) to obtain the residue. The residue was dissolved in deionized water (20 mL) to obtain the crude polyphenol extract of mutton patties. The crude polyphenol extract (0.5 mL) was added with 0.5 mL of Trichloroacetic acid solution (20% *w*/*v*). The mixture was swirled for 5 min and then centrifuged for 10 min (10,000 rpm and 4 °C) to obtain the supernatant. The supernatant was used to determine the total phenol content in the mutton patties.

### 2.6. Lipid Oxidation

The content of thiobarbituric acid reactive substances (TBARSs) was measured to determine the degree of lipid oxidation [2]. A standard curve was drawn using 1, 1, 3, 3-tetraethoxypropane as the standard for malondialdehyde (MDA). TBARS levels were expressed as mg MDA/kg sample.

### 2.7. Protein Oxidation

Protein carbonyl content was measured according to the method described by Xu et al. [14]. The standard curve was drawn with bovine serum albumin (BSA) as the standard, and a standard curve determined the protein concentration in the sample. The protein carbonyl content is expressed in nmol/mg protein with an extinction factor of 21.0/(mM·cm).

The content of protein sulfhydryl was determined using the methods of Shi et al. [15]. A total of 4.5 mL of Tris-glycine buffer (8 mol/L urea, pH 8.0) was mixed with 0.5 mL of myofibrillar protein solution (4 mg/mL). Next, 0.5 mL of Tris-HCl reagent (0.2 mol/L, pH 8.0) was added to the protein solution sample and reacted for 25 min (at 40 °C). The protein sulfhydryl content was measured using the absorbance at 412 nm. The extinction factor was 13.6/(mM·cm). The content of protein sulfhydryl is expressed in nmol/mg protein.

The cross-linking of proteins was assessed using 5% and 12% separating gels with SDS-PAGE (sodium dodecyl sulfate–polyacrylamide gel electrophoresis), as described by Wang et al. [2]. Molecular weight markers of 200, 120, 100, 70, 50, 40, 30, 25, and 14 kDa were used as references. Quantity One software (version 4.67, Bio-Rad, Hercules, CA, USA) was used to measure the relative optical densities of the gel bands.

### 2.8. Color

A colorimeter (SD9011, Chengdu Yike Instrument Equipment Co., Ltd., Chengdu, China) was used to measure the colors of the samples. The color of each sample was measured at three different positions, and the brightness (L*), red (a*), and yellow (b*) values were recorded.

### 2.9. Texture

The textures of the samples were measured using a texture analyzer (TA-XT Plus, Stable Micro System, London, UK). The P/50 probe was selected. The parameters were set as follows: the velocity of the pre-test was 3 mm, the test speed was 4 mm, the velocity of the post-test was 4 mm, the compression ratio was set to 50%, the trigger force was set to 5 g, and the interval of the test was 4 s. The hardness, chewiness, springiness, and cohesiveness of the samples were determined.

### 2.10. Microbiological Analysis

Microbiological analysis of the samples was conducted using the plate-counting method. In brief, 5 g of the samples and 50 mL of sterile saline solution (0.85%) were homogenized for 2 min in a sterile homogenization bag and then incubated at 4 °C for 20 min. Subsequently, 1:10 (*v*/*v*) dilutions were prepared. The total bacterial count was determined on agar plates and incubated for 48 h at 37 °C. The results are expressed in CFU/g.

### 2.11. Sensory Evaluation

A panel of eight experienced team members from Shihezi University assessed the color, juiciness, odor, taste, and overall acceptability of the mutton patties after 15 days of storage. The mutton patties were reheated in a microwave (901W; TG025QJG-NAH, Midea Group Co., Ltd., Guangdong, China) for 1 min and then cooled. The sensory characteristics were evaluated using a 7-point hedonic scale, as described by Wang et al. [2], where 7 represented ‘extremely liked’ and 1 represented ‘extremely disliked’. The results are reported as average values.

### 2.12. Statistical Analysis

All tests were conducted in triplicate (*n* = 3). One-way ANOVA was carried out using SPSS 19.0 (SPSS Inc., Chicago, IL, USA). Principal component analysis (PCA) was performed using Uncrambler version 9.7 (Camo ASA, Oslo, Norway). A significance level of *p* < 0.05 was used to determine statistical significance.

## 3. Results

### 3.1. Effect of TEO and Gum Arabic Ratio on the Embedding Efficiency

Embedding efficiency serves as an indicator of the wall material’s ability to encapsulate the core within the microcapsule, with a higher embedding efficiency indicating better core retention within the wall material [14]. As shown in Table 1, the core/wall ratio ranged from 1:2 to 1:12, and the embedding efficiency of both the TEO and total phenolic compounds (TPC) initially increased and then decreased (*p* < 0.05). When the core/wall ratio reached 1:8, the highest TEO embedding efficiency (90.32 ± 1.49%) and TPC embedding efficiency (92.13 ± 1.96%) were achieved, which were attributed to two main factors: excessive core material weakened the gum Arabic’s strength, resulting in a reduced embedding efficiency [14], while excessive wall material led to a highly viscous solution, impeding the diffusibility of the TEO and causing a decrease in the embedding efficiency [11]. Therefore, TEOMs with a core-to-wall ratio of 1:8 were selected for subsequent experiments. It is worth noting that gum Arabic demonstrated its effectiveness as a film-forming agent for TEO. Similarly, Xu et al. [14] used gum Arabic to encapsulate mulberry polyphenols, achieving high total phenol embedding efficiencies ranging from 88.34% to 95.54%.

### 3.2. The Stability of Phenolic Compounds in TEO and TEOMs in the Simulation System

To investigate the stability of phenols in the TEO and TEOMs, the present study simulated the processing conditions of mutton patties. As shown in Figure 1A, the phenolic compounds in both the TEO and TEOMs experienced a significant decrease during heating and storage. Notably, the loss in phenolic compounds in the TEOMs (21.12 ± 2.10%) was significantly lower than that in the TEO (33.31 ± 1.32%) after the heat treatment (*p* < 0.05). This outcome suggests that gum Arabic enhanced the stability of the phenolic compounds in essential oil by minimizing contact with heat, light, oxygen, and other factors [16]. Mahdavee-Khazaei et al. [17] reported that light, oxygen, and heat are likely key contributors to phenol degradation. Similarly, Felix et al. [18] demonstrated that gum Arabic, maltodextrin, and whey protein can enhance the thermal stability of cinnamon essential oil, elevating its degradation temperature. During storage after heating, the loss in phenolic compounds in the TEOMs increased to 32.12% on the 15th day. In comparison, the loss in phenolic compounds was 34.70% lower in the TEOMs than in the TEO. These results imply that gum Arabic acts as an effective barrier, delaying the degradation of phenols in TEO.

### 3.3. Stability of Phenolic Compounds in Mutton Patties

The total phenol content exhibited a notable decrease during the preparation and storage of mutton patties in all groups (Figure 1B, *p* < 0.05), indicating the degradation of phenolic substances. Similar findings were reported in other studies. Xu et al. [14] observed significant degradation of phenols, anthocyanins, and flavones in dried pork slices during processing and storage. On the 15th day, mutton patties treated with TEOMs displayed the highest total phenol content (0.49 ± 0.05 g GAE/kg sample), followed by the TEO + TEOMs treatment group (0.41 ± 0.03 g GAE/kg sample) and the TEO treatment group (0.32 ± 0.04 g GAE/kg sample). This suggests that gum Arabic is crucial in safeguarding against phenol loss. Importantly, before heat treatment, the total phenol content in mutton patties across the different treatments was lower than the added amount, with no significant difference between the three groups (*p* > 0.05). Le-Bourvellec and Renard [19] noted that phenolic substances can form covalent bonds with macromolecules in food (proteins, lipids, carbohydrates, etc.), reducing the antioxidant activity as covalent bonds increase in number. Therefore, aside from oxidation reactions, phenolic compounds may react with macromolecular components in mutton.

### 3.4. Effects of Different Treatments on Lipid Oxidation in Mutton Patties

The TBARS values of lamb patties were significantly increased during heat treatment and storage (*p* < 0.05, Figure 2A), indicating ongoing lipid oxidative damage. At all sampling time points, the control group consistently exhibited the highest TBARS values among all samples, while the TEO group had significantly lower TBARS values compared with the control (*p* < 0.05), indicating that phenolic compounds (e.g., carvacrol, thymol, and paracymene) in the TEO played a pivotal role in delaying lipid oxidation [5]. Kanatt et al. [20] reported that phenolic compounds possess metal ion-chelating abilities, particularly with ions like Fe^2+^, Zn^2+^, and Cu^2+^, which inhibit free radical formation, thus disrupting free radical reactions [21]. Furthermore, phenolic substances can serve as hydrogen atom donors for peroxyl radicals, generating phenoxyl radicals that further interact with other peroxyl radicals, preventing the oxidation of fats and proteins [22].

Across all sampling times, the TEOMs group consistently exhibited the lowest TBARS values among all the groups. Remarkably, on day 15, the TBARS value in the TEOM-treated mutton patties was 2.34 ± 0.12 mg MDA/kg, which was significantly lower than that in the control group (3.89 ± 0.12 mg MDA/kg), the TEO treatment group (3.42 ± 0.13 mg MDA/kg), and the TEO + TEOMs treatment group (2.78 ± 0.10 mg MDA/kg) (*p* < 0.05). This indicated that microencapsulation effectively enhanced the anti-lipid oxidation properties of the TEO. In their amorphous physical state, TEOMs have an increased contact area with the raw material compared with TEO, thus improving their protective effect on mutton patties [23]. Similar results reported that mold-resistant cinnamon–cranberry composite essential oil microcapsules mitigated the onset of lipid oxidation by reducing the oleic acid value of peanut kernels during storage [24]. In addition, Trommer and Neubert reported the dose-dependent protective effect of gum Arabic against lipid oxidation in simulated lipid systems [25]. Therefore, it is highly conceivable that gum Arabic in TEOMs exerts an inhibitory effect on lipid oxidation. Consistent with our findings, cinnamon oil microencapsulation was shown to slow the release of volatile antioxidant components, contributing to the delay of lipid oxidation in pork [26]. Modified hydrocolloid gum Arabic displayed a good antioxidant effect on fried potato strips [27]. Furthermore, lemon seed oil microcapsules demonstrated greater efficacy in inhibiting fat oxidation in beef jerky than lemon seed oil alone [28].

### 3.5. Effects of Different Treatments on Protein Oxidation in Mutton Patties

#### 3.5.1. Carbonyl Content

Protein carbonyl is an oxidation product of lysine, histidine, and arginine, serving as a reliable indicator of the protein oxidation degree [29]. As shown in Figure 2B, before the heat treatment, the carbonyl content did not significantly differ between the groups (*p* > 0.05). However, during the heat treatment and storage, the carbonyl content increased significantly in all mutton patty groups (*p* < 0.05), indicating that protein oxidative damage occurred, in accordance with the findings of Ganhão et al. [30], who observed a significant increase in the carbonyl content of beef burger patties during cooking and subsequent refrigeration, emphasizing the accelerating effect of heat treatment on protein oxidation.

The rate of carbonyl content increase in the TEO, TEO + TEOMs, and TEOMs groups was significantly lower than in the control group during heat treatment and storage (*p* < 0.05). On day 15, the carbonyl content in TEOM-treated mutton patties was 3.21 ± 0.19 nmol/mg protein, significantly lower than in the control group (5.15 ± 0.28 nmol/mg protein), TEO group (4.12 ± 0.21 nmol/mg protein), and TEO + TEOMs group (3.56 ± 0.17 nmol/mg protein) (*p* < 0.05), indicating the TEOMs’ superior inhibitory effect on protein oxidation in the mutton patties. Transition metals and lipid oxidation are known initiators of protein oxidation [31]. The metal-catalyzed formation of reactive oxygen species, which are responsible for attacking amino acid side chains, reportedly leads to carbonyl compound accumulation [21]. Lipid oxidation products (e.g., free radicals, hydroperoxides, MDA) can react with proteins, causing a loss of functional properties [15]. Hence, TEOMs protected against protein oxidation by inhibiting fat oxidation and chelating metal ions, which is consistent with the findings of Song et al. [32], who found that procyanidin microcapsules significantly inhibited protein oxidation in chicken sausages. The positive effects of vegetable extracts on meat products were also reported. Extracts from blackberry, dog rose, strawberry, and hawthorn effectively reduced the carbonyl compound accumulation in burger patties during cooking and storage [30]. Pomegranate peel extract can inhibit the protein oxidation of beef meatballs during the chilling period [21].

#### 3.5.2. Sulfhydryl Content

Sulfhydryl groups in cysteine can oxidize to form disulfide bonds, making sulfhydryl content an additional indicator of protein oxidation [15]. After the heat treatment, the protein sulfhydryl content significantly decreased in all the mutton patty groups (*p* < 0.05) (Figure 2C), reflecting the accelerated protein oxidation due to heating. This outcome is consistent with the findings of Xu et al. [14]. During storage, the sulfhydryl content significantly decreased in all groups (*p* < 0.05). At all sampling times, the sulfhydryl content in the TEOMs group was the highest among all the groups (*p* < 0.05). On day 15, the sulfhydryl content in the TEOM-treated mutton patties was 55.76 ± 0.69 nmol/mg protein, which was significantly higher than in the TEO + TEOMs group (50.27 ± 0.27 nmol/mg protein), TEO group (45.46 ± 0.76 nmol/mg protein), and control group (32.12 ± 1.38 nmol/mg protein) (all *p* < 0.05), suggesting that the TEOMs effectively reduced the sulfhydryl loss compared with the TEO. The TEOMs’ higher antioxidant activity may be attributed to the slow release of TEO, reducing its volatility [30]. In addition, Mariod reported that gum Arabic possesses antioxidant properties that scavenge free radicals and inactivate excited electrons [33]. Therefore, it is inferred that gum Arabic and TEO work synergistically to inhibit protein oxidation. In contrast, Jongberg et al. found that white grape seed extract accelerates sulfhydryl group loss during beef cooling [34]. Zhang et al. discovered that sage significantly reduces the sulfhydryl content in sausages during refrigerated storage [35]. The discrepancy in findings may stem from variations in protein conformation and properties and differences in phenolic compound chemical structures [36].

#### 3.5.3. Protein Cross-Linking

Protein oxidation can generate protein cross-links via disulfide and non-disulfide bonds, leading to protein aggregate accumulation [37]. SDS-PAGE was used to evaluate the myofibrillar protein oxidation in mutton patties (Figure 2D). Compared with before the heat treatment, the intensity of the actin and myosin bands significantly decreased in all treatment groups (*p* < 0.05) (Figure 2D’), indicating protein loss due to polymerization reactions [2]. These results were consistent with Turgut et al. [21], who found that myosin and actin are prone to sulfhydryl oxidation, resulting in disulfide-linked protein polymer formation. On day 15, the TEO, TEO + TEOMs, and TEOMs groups exhibited 99.63%, 76.36%, and 39.94% increased myosin band intensity and 63.19%, 47.76%, and 25.65% increased actin band intensity, respectively, compared with the control group. Therefore, the TEOMs displayed the strongest inhibitory effect on the protein oxidative cross-linking, which was attributed to the gradual release of bioactive compounds from the TEOMs in the mutton patties [10]. Song et al. similarly found that proanthocyanidin microencapsulation significantly inhibited disulfide bond formation in chicken sausages during storage [38]. Moreover, compared with before the heat treatment, high-molecular-weight protein aggregates appeared on the top of gels in all treatment groups on day 15, where they were unable to penetrate the gel grid. The accumulation of protein aggregates in the three treatment groups significantly decreased compared with the control group (*p* < 0.05), with the TEOMs treatment group showing the least accumulation (Figure 2D’). This phenomenon further indicates that TEOMs provide stronger protection against protein oxidation than TEO.

### 3.6. Effects of Different Treatments on the Color of Mutton Patties

Color is a crucial appearance characteristic of meat products that directly affects consumer acceptance. Table 2 presents the color changes in mutton patties for each group. Throughout the heat treatment and storage, there were no differences in b* values between the groups (*p* > 0.05). Similarly, blue pea flower petal extract was found to have no significant impact on the b* value of pork patties [39]. However, the L* and a* values significantly decreased in all groups (*p* < 0.05), which was mainly attributed to the interaction between lipid oxidation metabolites (e.g., hydrogen peroxide, MDA, and reactive oxygen species) and myoglobin in mutton, resulting in discoloration [2]. The reduction rate of L* and a* values in the TEO, TEO + TEOMs, and TEOMs groups was significantly lower than that in the control group (*p* < 0.05). On day 15, the TEOMs group exhibited the highest L* and a* values compared with the other groups. Thus, the microencapsulation of TEO with gum Arabic effectively improved the color protection of mutton patties, mainly due to the slow release of phenolic substances by the TEOMs [10]. Ghaderi-Ghahfarokhi et al. [40] reported similar findings, noting that TEOMs significantly inhibited the decrease in a* values in beef burgers during cold storage. Furthermore, Xu et al. [14] found that mulberry polyphenol microcapsules significantly inhibited the decrease in L* and a* values in dried minced pork slices during hot processing and cold storage.

### 3.7. Effects of Different Treatments on the Texture of Mutton Patties

The texture changes in mutton patties for each group are presented in Table 2. Throughout the heat treatment and storage, the cohesiveness and springiness did not differ between the groups (*p* > 0.05). However, the hardness significantly increased and the chewiness significantly decreased with time in all groups (*p* < 0.05). Notably, at each sampling point, the hardnesses of the TEO, TEO + TEOMs, and TEOMs groups were lower than those of the control group and the chewinesses of the TEO, TEO + TEOMs, and TEOMs groups were higher than those of the control group (*p* < 0.05). On the 15th day, the TEOMs group exhibited the lowest hardness and the highest chewiness, followed by the TEO + TEOMs and TEO groups. This suggests that microencapsulation significantly improved the TEO’s ability to suppress quality deterioration in the mutton patties.

It was established that hardness and chewiness influence consumer acceptability [30]. Similar findings were reported by Sun et al. [41], who found that microencapsulation of fennel essential oil significantly improved the chewability of minced pork. Lipid oxidation compromises the integrity of muscle fiber membranes, increasing product hardness [2]. Additionally, protein oxidation leads to protein cross-linked aggregate formation, which results in water separation from protein and increased hardness [30]. Furthermore, gum Arabic has hygroscopic properties, adsorbing water in the sample onto the microcapsule surface, reducing water loss [18]. Therefore, gum Arabic and TEO inhibit hardness deterioration by suppressing lipid and protein oxidation and absorbing moisture.

### 3.8. Effects of Different Treatments on the Microbiology of Mutton Patties

Table 2 summarizes the changes in total bacterial count in mutton patties. After the heat treatment, the total bacterial count significantly decreased in all groups (*p* < 0.05), mainly due to the sterilization from the high-temperature treatment, consistent with the observations of Xu et al. [42]. During storage, the bacterial count significantly increased (*p* < 0.05), with lower total bacterial counts in the TEO, TEO + TEOMs, and TEOMs groups compared with the control group (*p* < 0.05). On day 15, the TEOMs group had a significantly lower total bacterial count than the other groups, indicating that the TEO microencapsulation had a stronger antibacterial effect. Phenolic compounds can penetrate bacterial cell membranes’ phospholipid bilayers, disrupting bacterial enzyme systems and inhibiting bacterial growth [36]. In addition, Cai et al. [11] reported that microcapsules effectively enhance essential oil diffusion in the medium, damaging cell membrane integrity and causing bacterial death. Similar findings were reported by Hu et al. [26], who found that the microencapsulation of cinnamon oil significantly improves the antibacterial effects in chilled pork. Moreover, Wang et al. [43] found that clove essential oil microcapsules applied in meat products can effectively inhibit mold growth and have excellent antifungal effects.

### 3.9. Sensory Evaluation

After 15 days of storage, the sensory properties of mutton patties were evaluated, including color, juiciness, odor, taste, and overall acceptability (Table 3). The TEOMs treatment group had a significantly higher color score than the other treatment groups (*p* < 0.05), mainly due to its higher L* and a* values compared with the other groups. The juiciness score was the highest in the TEOMs treatment group, which was attributed to the antioxidant and hygroscopic properties of the microcapsules. For the odor and taste parameters, the scores of the TEO, TEO + TEOMs, and TEOMs groups were higher than those of the control group, with the TEOMs group scoring significantly higher than the TEO group (*p* < 0.05), indicating that the microencapsulation significantly reduced the unique odor of TEO. Regarding the overall acceptability, the TEOMs group received the highest score, implying that the TEOMs effectively improved the quality characteristics of the mutton patties. The result of this study was consistent with those reported by Li et al. [44], who found that the addition of clove essential oil microcapsules to smoked horsemeat sausages significantly improved the organoleptic indices of color, odor, and flavor compared with the control group.

### 3.10. Principal Component Analysis

PCA was conducted to determine the quality parameters affected by the TEOMs in the mutton patties after 15 days of storage. The PCA results revealed that the first two principal components accounted for 99.22% of the total variability (PC1: 84.12%, PC2: 15.10%) (Figure 3A). Projections of different treatments in two-dimensional space with PC1 and PC2 as loading factors are depicted in Figure 3B. The control group was positioned on the positive side of PC1, which was closely related to the TBARS value, carbonyl value, total bacterial count, hardness, and chewiness, indicating higher levels of these variables in the control group. Conversely, the TEO, TEO + TEOMs, and TEOMs groups were situated on the negative side of PC1, suggesting lower levels of these parameters in the three treatment groups. Compared with the TEO-treated mutton patties, those treated with TEO + TEOMs and TEOMs were closer in terms of sensory characteristics (color, juiciness, odor, taste, and overall acceptability). This indicates that microcapsules improved the sensory quality of mutton patties. Furthermore, the TEOMs group was further from the fat oxidation and protein oxidation indicators than the TEO + TEOMs and TEO groups, highlighting that microencapsulation was the most efficient way to delay oxidation reactions and improve the quality of mutton patties.

## 4. Conclusions

This study demonstrated that gum Arabic was a suitable encapsulating agent for the TEO. A core/wall ratio of 1:8 exhibited the highest encapsulation efficiency (90.32 ± 1.49%) for the TEO. The microencapsulation treatment improved the stability of the TEO phenolics and the slow release of the TEO improved the antioxidant and antimicrobial activities of the mutton patties during storage. On the 15th day, compared with the control, TEO, TEO + TEOMs groups, the TEOMs group displayed reduced thiobarbituric acid content, carbonyl content, loss of sulfhydryl groups, and protein cross-linking in the mutton patties. In addition, the TEOMs positively affected the color, texture, microbial stability, and organoleptic properties of the mutton patties. Therefore, the microencapsulation of the TEO effectively enhanced the quality and safety of the mutton patties.

## Figures and Tables

**Figure 1 foods-12-03758-f001:**
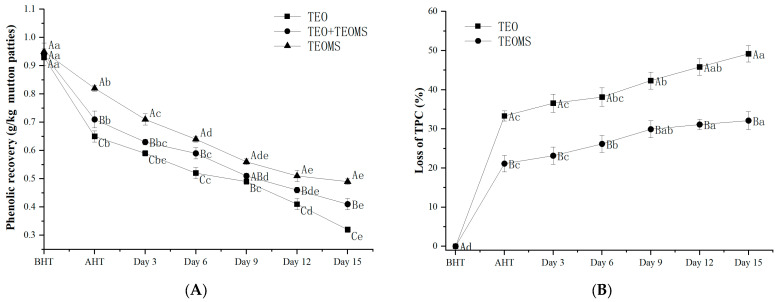
(**A**) Stability of phenolic compounds in TEO and TEOMs in the simulation system; (**B**) stability of phenolic compounds in mutton patties. TEO: thyme essential oil; TEOMs: thyme essential oil microcapsules; BHT: before heat treatment; AHT: after heat treatment. Values with different letters (A–C) at the same sampling time were significantly different (*p* < 0.05). Values with different letters (a–e) in the same batch were significantly different (*p* < 0.05).

**Figure 2 foods-12-03758-f002:**
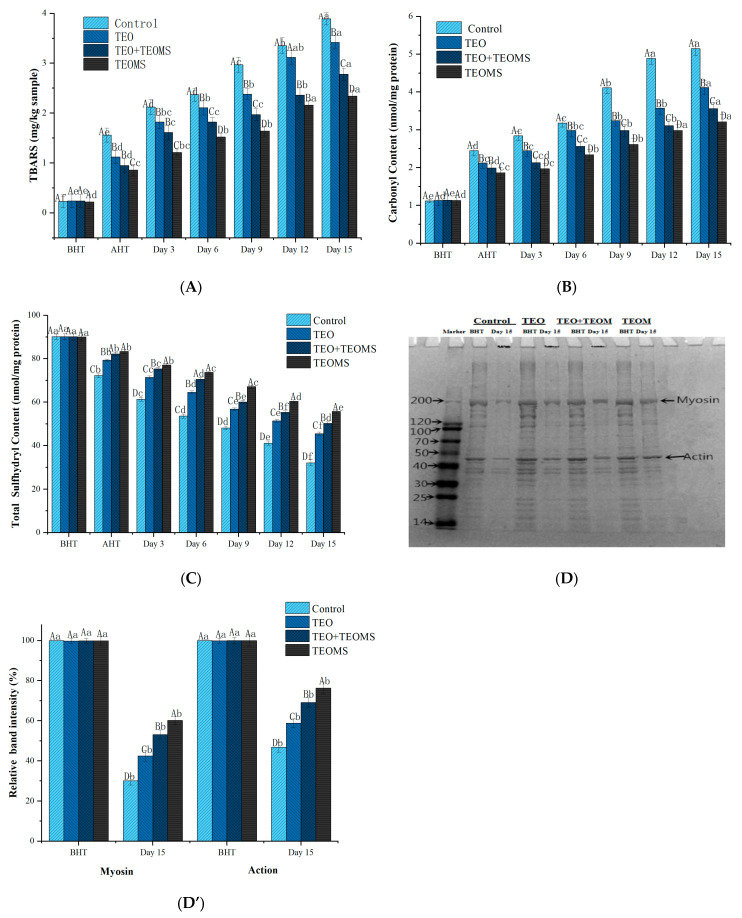
(**A**) Effects of different treatments on TBARS values in mutton patties; (**B**) effects of different treatments on carbonyl content in mutton patties; (**C**) effects of different treatments on total sulfhydryl content in mutton patties; (**D**) representative SDS-PAGE gel of myofibrillar protein; (**D’**) analysis of relative optical density intensity of myosin and action bands. TEO: thyme essential oil; TEOMs: thyme essential oil microcapsules; BHT: before heat treatment; AHT: after heat treatment. Values with different letters (A–D) at the same sampling time were significantly different (*p* < 0.05). Values with different letters (a–f) in the same batch were significantly different (*p* < 0.05).

**Figure 3 foods-12-03758-f003:**
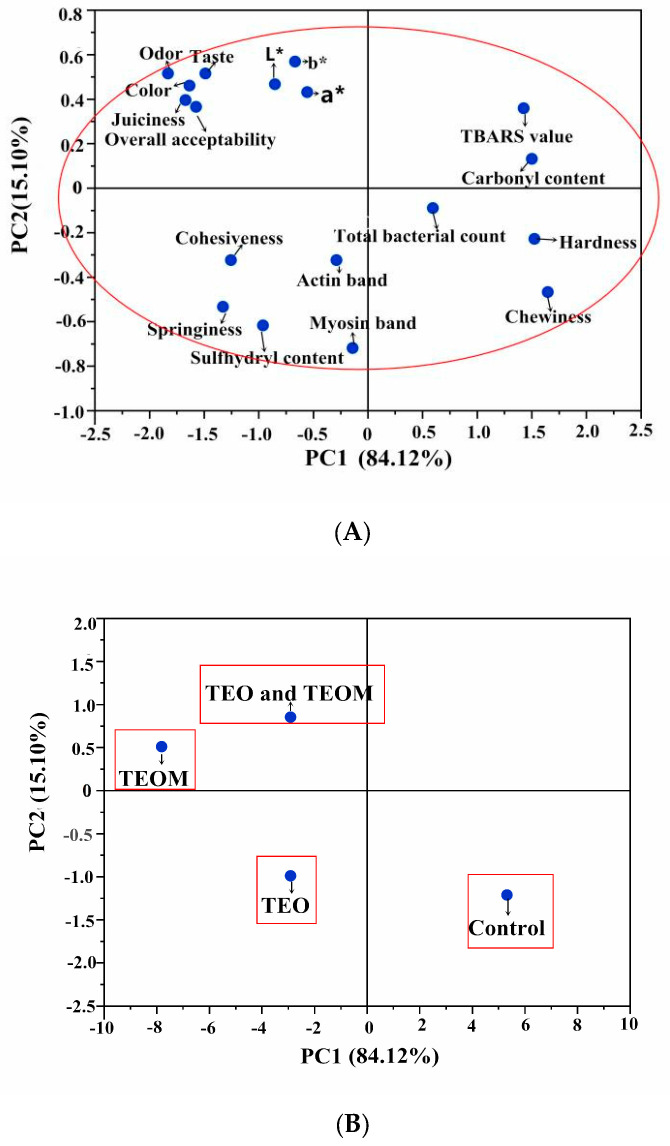
Principal component analysis (PCA) for lipid oxidation, protein oxidation, microbial, color, and sensory qualities of mutton patties after storage for 15 days: (**A**) loadings for the first two principal components; (**B**) weighed PCA bi-plot of scores. TEO: thyme essential oil; TEOMs: thyme essential oil microcapsules; *: the difference is significant.

**Table 1 foods-12-03758-t001:** Effect of TEO and gum Arabic ratio on the embedding efficiency of TEO and total phenol.

Ration (*w*/*w*)	TPC in Microcapsules (mg GAE/g)	EE_TPC_ (%)	EE_TEO_ (%)
1:2	128.54 ± 2.32 ^a^	62.34 ± 1.12 ^e^	60.36 ± 1.39 ^d^
1:4	79.21 ± 1.61 ^b^	76.83 ± 0.92 ^d^	74.32 ± 1.27 ^c^
1:6	58.60 ± 1.59 ^c^	85.26 ± 1.32 ^b^	86.13 ± 1.84 ^b^
1:8	47.49 ± 1.83 ^d^	92.13 ± 1.96 ^a^	90.32 ± 1.49 ^a^
1:10	34.44 ± 1.72 ^e^	83.52 ± 1.79 ^bc^	86.12 ± 1.38 ^b^
1:12	28.13 ± 1.16 ^f^	81.87 ± 1.66 ^c^	85.32 ± 1.47 ^b^

Note: Mean values bearing different superscripts (small letters in the same column) indicate significant differences (*p* < 0.05, *n* = 3). TPC: total phenol content; TEO: thyme essential oil; EE: embedding efficiency.

**Table 2 foods-12-03758-t002:** Changes in color parameters, texture properties, and microbiological stability of mutton patties.

Variables	Batches	Days of Storage						
		BHT	AHT	Day 3	Day 6	Day 9	Day 12	Day 15
L*	Control	46.13 ± 1.41 ^Aa^	35.24 ± 1.38 ^Db^	33.42 ± 1.83 ^Db^	30.61 ± 1.41 ^Dc^	28.17 ± 1.44 ^Dc^	27.34 ± 1.19 ^Dc^	24.23 ± 1.41 ^Dd^
	TEO	46.16 ± 1.34 ^Aa^	37.61 ± 1.17 ^Cb^	35.83 ± 1.19 ^Cb^	33.39 ± 1.38 ^Cc^	30.72 ± 1.21 ^Ccd^	28.17 ± 1.27 ^Cd^	27.54 ± 1.76 ^Cd^
	TEO + TEOMs	46.21 ± 1.52 ^Aa^	40.39 ± 1.24 ^Bb^	38.12 ± 1.42 ^Bb^	35.17 ± 1.82 ^Bc^	33.69 ± 1.43 ^Bcd^	30.56 ± 1.24 ^Bd^	29.14 ± 1.63 ^Bd^
	TEOMs	46.23 ± 1.63 ^Aa^	43.45 ± 1.31 ^Aa^	40.51 ± 1.65 ^Ab^	38.32 ± 1.45 ^Ab^	35.13 ± 1.32 ^Ac^	33.42 ± 1.14 ^Ac^	31.67 ± 1.42 ^Ad^
a*	Control	10.61 ± 0.23 ^Aa^	8.18 ± 0.24 ^Cb^	7.93 ± 0.22 ^Db^	7.45 ± 0.18 ^Dc^	6.84 ± 0.16 ^Dd^	5.95 ± 0.13 ^De^	5.13 ± 0.19 ^Df^
	TEO	10.63 ± 0.18 ^Aa^	8.97 ± 0.27 ^Bb^	8.12 ± 0.23 ^Cc^	8.07 ± 0.21 ^Cc^	7.03 ± 0.13 ^Cd^	6.23 ± 0.27 ^Ce^	5.78 ± 0.23 ^Cf^
	TEO + TEOMs	10.62 ± 0.13 ^Aa^	9.12 ± 0.15 ^Bb^	8.56 ± 0.21 ^Bc^	8.32 ± 0.19 ^Bc^	7.56 ± 0.28 ^Bd^	6.78 ± 0.26 ^Be^	6.16 ± 0.27 ^Bf^
	TEOMs	10.61 ± 0.17 ^Aa^	9.67 ± 0.17 ^Ab^	9.17 ± 0.27 ^Ac^	8.75 ± 0.26 ^Ac^	8.13 ± 0.29 ^Ad^	7.56 ± 0.25 ^Ae^	7.14 ± 0.28 ^Ae^
b*	Control	14.43 ± 0.51 ^Aa^	12.13 ± 0.52 ^Ab^	11.43 ± 0.47 ^Ab^	10.65 ± 0.54 ^Ac^	9.98 ± 0.53 ^Ac^	9.65 ± 0.56 ^Ac^	8.56 ± 0.52 ^Ad^
	TEO	14.41 ± 0.49 ^Aa^	12.14 ± 0.51 ^Ab^	11.45 ± 0.31 ^Ab^	10.67 ± 0.49 ^Ac^	9.97 ± 0.37 ^Ac^	9.67 ± 0.47 ^Ac^	8.57 ± 0.47 ^Ad^
	TEO + TEOMs	14.42 ± 0.47 ^Aa^	12.16 ± 0.49 ^Ab^	11.46 ± 0.53 ^Ab^	10.68 ± 0.38 ^Ac^	9.97 ± 0.31 ^Ac^	9.68 ± 0.43 ^Ac^	8.58 ± 0.48 ^Ad^
	TEOMs	14.42 ± 0.46 ^Aa^	12.21 ± 0.38 ^Ab^	11.48 ± 0.52 ^Ab^	10.69 ± 0.41 ^Ac^	9.98 ± 0.47 ^Ac^	9.68 ± 0.42 ^Ac^	8.58 ± 0.49 ^Ad^
Hardness (N)	Control	54.61 ± 1.31 ^Af^	70.43 ± 1.49 ^Ae^	75.78 ± 1.76 ^Ade^	81.23 ± 1.65 ^Ad^	86.46 ± 1.38 ^Ac^	91.53 ± 1.76 ^Ab^	97.12 ± 1.34 ^Aa^
	TEO	53.18 ± 1.53 ^Ae^	65.19 ± 1.33 ^Bd^	70.34 ± 1.81 ^Bc^	77.43 ± 1.41 ^Bb^	80.12 ± 1.97 ^Bb^	86.59 ± 1.59 ^Ba^	90.69 ± 1.75 ^Ba^
	TEO + TEOMs	53.40 ± 1.51 ^Ae^	63.62 ± 1.41 ^Bd^	66.78 ± 1.85 ^Cd^	70.52 ± 1.34 ^Cc^	75.59 ± 1.94 ^Cb^	79.91 ± 1.74 ^Cb^	84.17 ± 1.98 ^Ca^
	TEOMs	54.11 ± 1.54 ^Ad^	60.12 ± 1.65 ^Cc^	63.32 ± 1.75 ^Dc^	65.31 ± 1.35 ^Dc^	68.18 ± 1.87 ^Dc^	73.45 ± 1.59 ^Db^	78.12 ± 1.84 ^Da^
Springiness	Control	0.45 ± 0.01 ^Aa^	0.41 ± 0.02 ^Aab^	0.38 ± 0.01 ^Ab^	0.33 ± 0.03 ^Ab^	0.29 ± 0.03 ^Ac^	0.26 ± 0.06 ^Ac^	0.23 ± 0.07 ^Ac^
	TEO	0.46 ± 0.02 ^Aa^	0.42 ± 0.01 ^Aa^	0.39 ± 0.02 ^Aa^	0.33 ± 0.01 ^Ab^	0.28 ± 0.01 ^Ab^	0.27 ± 0.04 ^Ab^	0.24 ± 0.03 ^Ac^
	TEO + TEOMs	0.47 ± 0.02 ^Aa^	0.41 ± 0.03 ^Ab^	0.40 ± 0.04 ^Ab^	0.34 ± 0.02 ^Ac^	0.29 ± 0.04 ^Ac^	0.27 ± 0.02 ^Ac^	0.23 ± 0.04 ^Ad^
	TEOMs	0.47 ± 0.03 ^Aa^	0.42 ± 0.01 ^Aa^	0.40 ± 0.05 ^Aa^	0.34 ± 0.04 ^Ab^	0.30 ± 0.05 ^Ab^	0.28 ± 0.03 ^Abc^	0.25 ± 0.07 ^Ac^
Cohesiveness	Control	0.41 ± 0.03 ^Aa^	0.38 ± 0.02 ^Aa^	0.35 ± 0.01 ^Ab^	0.32 ± 0.01 ^Ab^	0.28 ± 0.02 ^Ac^	0.27 ± 0.03 ^Ac^	0.21 ± 0.05 ^Ac^
	TEO	0.42 ± 0.04 ^Aa^	0.37 ± 0.03 ^Aa^	0.36 ± 0.03 ^Ab^	0.32 ± 0.02 ^Ab^	0.28 ± 0.03 ^Ab^	0.28 ± 0.04 ^Ab^	0.22 ± 0.02 ^Ac^
	TEO + TEOMs	0.41 ± 0.02 ^Aa^	0.38 ± 0.02 ^Aa^	0.36 ± 0.02 ^Ab^	0.31 ± 0.03 ^Ab^	0.29 ± 0.04 ^Ab^	0.29 ± 0.01 ^Ab^	0.22 ± 0.01 ^Ac^
	TEOMs	0.43 ± 0.03 ^Aa^	0.39 ± 0.05 ^Aa^	0.37 ± 0.06 ^Aa^	0.33 ± 0.04 ^Ab^	0.30 ± 0.01 ^Ab^	0.29 ± 0.04 ^Ab^	0.23 ± 0.02 ^Ac^
Chewiness (N)	Control	22.31 ± 1.43 ^Aa^	18.43 ± 1.03 ^Cb^	16.56 ± 1.46 ^Dc^	14.31 ± 1.34 ^Dd^	13.56 ± 1.38 ^Dd^	12.17 ± 1.01 ^De^	11.42 ± 1.36 ^De^
	TEO	22.73 ± 1.31 ^Aa^	19.46 ± 1.34 ^Bb^	17.58 ± 1.18 ^Cc^	16.54 ± 1.31 ^Cc^	14.34 ± 1.27 ^Cd^	13.18 ± 1.18 ^Cd^	12.78 ± 1.39 ^Cd^
	TEO + TEOMs	22.52 ± 1.34 ^Aa^	19.98 ± 1.33 ^Bb^	18.61 ± 1.23 ^Bc^	17.34 ± 1.41 ^Bc^	16.52 ± 1.35 ^Bd^	15.45 ± 1.27 ^Bd^	14.83 ± 1.17 ^Bd^
	TEOMs	22.62 ± 1.42 ^Aa^	20.43 ± 1.21 ^Ab^	19.67 ± 1.31 ^Ab^	18.87 ± 1.671 ^Ab^	17.35 ± 1.30 ^Ac^	16.47 ± 1.33 ^Ac^	15.67 ± 1.35 ^Ac^
Total bacterial count (log CFU/g)	Control	4.63 ± 0.32 ^Ac^	3.48 ± 0.43 ^Ad^	3.76 ± 0.39 ^Ad^	4.45 ± 0.39 ^Ac^	5.34 ± 0.37 ^Ab^	5.74 ± 0.65 ^Aab^	6.03 ± 0.36 ^Aa^
	TEO	4.62 ± 0.34 ^Ab^	3.39 ± 0.51 ^Bd^	3.51 ± 0.41 ^Bcd^	3.96 ± 0.41 ^Bc^	4.57 ± 0.41 ^Bb^	4.87 ± 0.47 ^Bab^	5.49 ± 0.43 ^Ba^
	TEO + TEOMs	4.61 ± 0.31 ^Ab^	3.21 ± 0.36 ^Cc^	3.32 ± 0.42 ^Cc^	3.62 ± 0.34 ^Cc^	4.23 ± 0.35 ^Cb^	4.58 ± 0.42 ^Cb^	5.04 ± 0.49 ^Ca^
	TEOMs	4.62 ± 0.34 ^Aa^	3.19 ± 0.41 ^Cc^	3.21 ± 0.46 ^Dc^	3.38 ± 0.39 ^Dc^	3.96 ± 0.67 ^Db^	4.32 ± 0.59 ^Db^	4.77 ± 0.59 ^Da^

Note: Mean values bearing different superscripts (small letters in the same row and capital letters in the same column) indicate significant differences (*p* < 0.05, *n* = 3). TEO: thyme essential oil; TEOMs: thyme essential oil microcapsules; BHT: before heat treatment; AHT: after heat treatment.

**Table 3 foods-12-03758-t003:** Sensory evaluations of mutton patties.

Treatments	Color	Juiciness	Odor	Taste	Overall Acceptability
Control	3.77 ± 0.91 ^D^	3.56 ± 1.13 ^D^	3.23 ± 1.17 ^D^	4.17 ± 1.12 ^D^	4.62 ± 0.87 ^D^
TEO	4.86 ± 0.83 ^C^	4.59 ± 0.93 ^C^	4.13 ± 0.93 ^C^	4.93 ± 1.03 ^C^	4.98 ± 1.01 ^C^
TEO + TEOMs	5.15 ± 0.72 ^B^	5.32 ± 1.12 ^B^	5.22 ± 0.94 ^B^	5.14 ± 0.95 ^B^	5.32 ± 1.14 ^B^
TEOMs	5.63 ± 0.84 ^A^	5.76 ± 1.04 ^A^	5.93 ± 1.17 ^A^	5.63 ± 1.13 ^A^	6.01 ± 1.03 ^A^

Note: Mean values bearing different superscripts (capital letters in the same column) indicate significant differences (*p* < 0.05, *n* = 3). TEO: thyme essential oil; TEOMs: thyme essential oil microcapsules.

## Data Availability

Data are contained within the article.

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
