# Peer review of "Effects of Thyme Essential Oil Microcapsules on the Antioxidant and Quality Characteristics of Mutton Patties"

_foods, 2023, doi:10.3390/foods12203758_

Round 1

Reviewer 1 Report

Manuscript foods-2646627, entitled “Effects of thyme essential oil microcapsules on the antioxidant and quality characteristics of mutton patties”

 This article provides useful information on the effects of thyme essential oil microcapsules on the antioxidant and quality characteristics of mutton patties. It is in general appropriately organized, carried out and written, however there are some points that should be corrected or clarified.

 My main concern is that you should remove several parts from results in the section of discussion (L199-205, 229-234, 243-245, 251-255, 283-289, 295-305, 313-317, 324-335, 341-342, 348-357, 365-367, 371-373, 436-440, 443-449, 476-483, 488-489, 493-499 etc)

In text, in references with more than 3 authors, the term “et al.” should be used. Please check L47, 121-122, 140-141, 204, 231-232, 243, 284-285, 314, 342, 365, 445, 447, 489, 495, 497 etc. Especially in L330, check reference style.

 L145: “…was determined by the method of Shi et al. [11].”

L146: Please delete “as standard”

L146-147: Please rephrase

L182: “are reported”

L280: Fig. 2A or Fig. 2B?

L310: Fig. 2AB or Fig. 2A?

L470: Chewiness was significantly decreased with time

L472: Higher for chewiness

L473: “On the 15th day, the TEOM group exhibited the lowest hardness and the highest chewiness…”

L497-498: “…been reported by Hu et al. [21], who found that…”

L548: “further”

Minor editing of English language required

Author Response

Thanks for your valuable advice. Besides, I revised the paper according to editor and reviewer’s comments. We have added some related information into the manuscript according to your suggestion. The revised texts were highlighted with a red color text.

The issues raised by the reviewers in the comments are addressed below:

Comment 1. My main concern is that you should remove several parts from results in the section of discussion (L199-205, 229-234, 243-245, 251-255, 283-289, 295-305, 313-317, 324-335, 341-342, 348-357, 365-367, 371-373, 436-440, 443-449, 476-483, 488-489, 493-499 etc)

Answer: Thank you for your suggestions. For the“Results and Discussion”, we have the following thoughts: firstly, according to the journal related requirements results and discussion can be written as one chapter, secondly, writing the results and discussion sections together can help to ensure the flow and coherence of the paper. This approach would better demonstrate the progress and relevance of the study without introducing unnecessary separation between the two sections. But we will take care of the related issues in our future article writing.

Comment 2.In text, in references with more than 3 authors, the term et al. should be used. Please check L47, 121-122, 140-141, 204, 231-232, 243, 284-285, 314, 342, 365, 445, 447, 489, 495, 497 etc. Especially in L330, check reference style.

Answer: We are very sorry for our negligence. We have made changes to the reference format.

Comment 3.L145: “…was determined by the method of Shi et al. [11].

Answer: Thank you for your suggestions. As you suggested, now in our manuscript line 151, we have revised the manuscript.

The content of protein sulfhydryl was determined by the methods of Shi et al. [15].

Comment 4.L146: Please delete as standard

Answer: Thank you for your suggestions. As you suggested, now in our manuscript line 152-155, we have revised the manuscript.

4.5 mL of Tris-glycine buffer (8 mol/L urea, pH 8.0) was mixed with 0.5 mL of myofibrillar protein solution (4 mg/mL). Next, 0.5 mL of Tris-HCl reagent (0.2 mol/L, pH 8.0) was added to the protein solution sample and reacted for 25 min (at 40 ℃). The protein sulfhydryl content was measured by absorbance at 412 nm.

Comment 5.L146-147: Please rephrase

Answer: Thank you for your suggestions. As you suggested, now in our manuscript line 152-155, we have revised the Manuscript.

4.5 mL of Tris-glycine buffer (8 mol/L urea, pH 8.0) was mixed with 0.5 mL of myofibrillar protein solution (4 mg/mL). Next, 0.5 mL of Tris-HCl reagent (0.2 mol/L, pH 8.0) was added to the protein solution sample and reacted for 25 min (at 40 ℃). The protein sulfhydryl content was measured by absorbance at 412 nm.

Comment 6.L182: are reported

Answer: Thank you for your suggestions. As you suggested, now in our manuscript line 190, we have revised the Manuscript.

The results are reported as average values.

Comment 7.L280: Fig. 2A or Fig. 2B?

Answer: We are very sorry for our negligence. As you suggested, now in our manuscript line 278, We have reordered the pictures (L280: Fig. 2A).

Comment 8.L310: Fig. 2AB or Fig. 2A?

Answer: We are very sorry for our negligence. As you suggested, now in our manuscript line 311, We have reordered the pictures (L310: Fig. 2B).

Comment 9.L470: Chewiness was significantly decreased with time

Answer: We are very sorry for our negligence. As you suggested, now in our manuscript line 475-476, we have revised the Manuscript.

Hardness significantly increased and chewiness was significantly decreased with time in all groups (p < 0.05).

Comment 10.L472: Higher for chewiness

Answer: We are very sorry for our negligence. As you suggested, now in our manuscript line 476-479, we have revised the Manuscript. Notably, at each sampling point, the hardness of the TEO, TEO + TEOM, TEOM groups were lower than those of the control group and chewiness of the TEO, TEO + TEOM, TEOM groups were higher than those of the control group (p < 0.05).

Comment 11.L473: On the 15th day, the TEOM group exhibited the lowest hardness and the highest chewiness…”

Answer: Thank you for your suggestions. As you suggested, now in our manuscript line 479-480, we have revised the Manuscript.

On the 15th day, the TEOM group exhibited the lowest hardness and the highest chewiness, followed by the TEOM + TEO and TEO groups.

Comment 12.L497-498: “…been reported by Hu et al. [21], who found that…”

Answer: Thank you for your suggestions. As you suggested, now in our manuscript line 505, we have revised the Manuscript.

 Similar findings have been reported by Hu et al. [26], who found that microencapsulation of cinnamon oil significantly improved antibacterial effects in chilled pork.

Comment 13.L548: further

Answer: Thank you for your suggestions. As you suggested, now in our manuscript line 545, we have revised the Manuscript.

 Furthermore, the TEOM group was further from fat oxidation and protein oxidation indicators than the TEO + TEOM and TEO groups,

We tried our best to improve the manuscript and made some changes in the manuscript. We appreciate for Editor/Reviewer’s warm work earnestly and hope that the correction will meet with approval. Once again thank you very much for your comments and suggestions.

Best regards!

Sincerely yours,

Jingyun Wang

Reviewer 2 Report

Thank you for the opportunity to review the submitted Manuscript.

I have some major complaints:

The Introduction is too poor

- please expand it with the advantages of TEO, e.g.: antibacterial effect has been demonstrated against..., anti-inflammatory... etc.

- what is TEO added to in the food industry? please take this information into account

- The Authors mentioned microencapsulation - is it used in the case of TEO? In what cases? if not, please consider what other oils are encapsulated and used in the food industry

Formulas 1, 2 and 3 are illegible, please change them using a special program for this purpose

Why do the Authors sometimes use the full name "gram" and other times the abbreviation "g". Please harmonize this throughout the Manuscript

There is no need to use the full name "minutes". Please use the abbreviation and refer it to the entire Manuscript

Is method 2.2 an innovative method? invented by the Authors? or was microencapsulation carried out based on previous research? if so, please provide a citation

Tables 1 and 2. Please add legends including all abbreviations used in the tables. The table descriptions used by the authors do not comply with MDPI requirements

Figure 2 - please increase the distance between the groups of posts and change their filling. In its current form, this figure is unreadable

There was no discussion at work. During describing the results, the Authors try to refer to the results of other researchers, but this is relatively too little. Moreover, the "Results" chapter is not a "Results and discussion" chapter. Please expand the discussion within the results and change the chapter title or create a separate "Discussion" chapter.

Good luck

Author Response

Thanks for your valuable advice. Besides, I revised the paper according to editor and reviewer’s comments. We have added some related information into the manuscript according to your suggestion. The revised texts were highlighted with a red color text.

The issues raised by the reviewers in the comments are addressed below:

Comment 1. please expand it with the advantages of TEO, e.g.: antibacterial effect has been demonstrated against..., anti-inflammatory... etc.what is TEO added to in the food industry? please take this information into account

Answer: Thank you for your suggestions. As you suggested, now in our manuscript line 38-45, we have revised the Manuscript.

Thyme essential oil has good antimicrobial properties against a variety of microorganisms, such as: Listeria monocytogenes, Escherichia coli and Staphylococcus aureus [7]. Thyme essential oil is often used as a good antioxidant for food preservation. Ghafarifarsani et al. found that improving the antioxidant capacity of fish by adding thyme essential oil to fish feed improved the health of fish [8]. Zhang et al. found that the new blend of thyme essential oil (curdlan/polyvinyl alcohol film) extends the shelf life of frozen meat by 10 days [9].

Comment 2. The Authors mentioned microencapsulation - is it used in the case of TEO? In what cases? if not, please consider what other oils are encapsulated and used in the food industry

Answer: Thank you for your suggestion. Microencapsulation is used in the case of TEO. As you suggested, now in our manuscript line 52-56, we have revised the manuscript.

 Recent studies have shown that Thyme Essential Oil Microencapsulation (TEOM) technology can produce positive ameliorative effects in an in vitro mimetic system. For instance, Cai et al. [11], reported that microcapsule technology enhances TEO's stability and water solubility. Additionally, Zhang et al. [6] found that TEOM exhibits superior antibacterial properties compared to TEO.

Comment 3. Formulas 1, 2 and 3 are illegible, please change them using a special program for this purpose

Answer: We are very sorry for our negligence. We have modified the formulas 1, 2 and 3.

Comment 4.Why do the Authors sometimes use the full name "gram" and other times the abbreviation "g". Please harmonize this throughout the Manuscript

Answer: We are very sorry for our negligence. We have made changes to change the full name "gram" to "g".

Comment 5.There is no need to use the full name "minutes". Please use the abbreviation and refer it to the entire Manuscript

Answer: We are very sorry for our negligence. Now we use the abbreviation and refer it to the entire Manuscript instead the full name "minutes".

Comment 6.Is method 2.2 an innovative method? invented by the Authors? or was

microencapsulation carried out based on previous research? if so, please provide a

citation

Answer: Thank you for your suggestion. As you suggested, now in our manuscript line 73, we have revised the Manuscript.

The method of Devi , Das, and Badwaik was used with some modifications[12].

[12] Devi L M.; Das A.; Badwaik L. Effect of gelatin and acacia gum on anthocyanin coacervated microcapsules using double emulsion and its characterization. International journal of biological macromolecules. 2023, 123896.

Comment 7. Tables 1 and 2. Please add legends including all abbreviations used in the tables. The table descriptions used by the authors do not comply with MDPI requirements

Answer: Thank you for your suggestion. According your suggestion, we have revised the table and added the legends including all abbreviations used in the tables.

Comment 8. Figure 2 - please increase the distance between the groups of posts and change their filling. In its current form, this figure is unreadable

Answer: Thank you for your suggestion, we have now changed the fill between of different groups.

Comment 9. There was no discussion at work. During describing the results, the Authors try to refer to the results of other researchers, but this is relatively too little. Moreover, the "Results" chapter is not a "Results and discussion" chapter. Please expand the discussion within the results and change the chapter title or create a separate "Discussion" chapter.

Answer: Thank you for your suggestions. As you suggested, we have added findings from other researchers.

3.4. Effects of different treatments on lipid oxidation in mutton patties:

L295-298: Similar results have been reported that mould-resistant cinnamon-cranberry composite essential oil microcapsules mitigated the onset of lipid oxidation by reducing the oleic acid value of peanut kernels during storage [24].

3.7. Effects of different treatments on the texture of mutton patties:

L484-485: Similar findings were reported by Sun et al. [41], who found that microencapsulation of fennel essential oil significantly improved the chewability of minced pork.

3.8. Effects of different treatments on the microbiology of mutton patties

L506-508: Moreover, Wang et al. [43] found that clove essential oil microcapsules applied in meat products can effectively inhibit mold growth and have excellent antifungal effects.

3.9. Sensory evaluation

L521-525: The result of this study was consistent with those reported by Li et al. [44], who found that the addition of clove essential oil microcapsules to smoked horsemeat sausages significantly improved the organoleptic indices of colour, odour and flavour compared to the control group.

For the “Results and Discussion”, we have the following thoughts: firstly, according to the journal related requirements results and discussion can be written as one chapter, secondly, writing the results and discussion sections together can help to ensure the flow and coherence of the paper. This approach would better demonstrate the progress and relevance of the study without introducing unnecessary separation between the two sections.

We tried our best to improve the manuscript and made some changes in the manuscript. We appreciate for Editor/Reviewer’s warm work earnestly and hope that the correction will meet with approval. Once again thank you very much for your comments and suggestions.

Best regards!

Sincerely yours,

Jingyun Wang

Reviewer 3 Report

This study aimed to assess the stability of thyme essential oil microcapsules (TEOM) and their impact on the antioxidant properties and quality of lamb patties. The research investigates the impact of TEOM on various aspects, including thiobarbituric acid content, carbonyl content, sulfhydryl levels, cross-linking, color, texture, microbiological stability, and sensory attributes of mutton pies. The results demonstrated that gum arabic effectively enhanced the stability of phenols within thyme essential oil (TEO), with an optimal core/wall ratio of 1:8. Substituting TEO with TEOM in lamb patties led to reductions in thiobarbituric acid content, carbonyl content, sulfhydryl loss, and protein cross-linking. Additionally, TEOM positively influenced the mutton patties' color, texture, microbiological stability, and sensory attributes. This study demonstrates that gum arabic is a suitable encapsulating agent for TEO. A core/wall ratio of 1:8 exhibited the highest encapsulation efficiency for TEO. Microcapsules significantly improved the stability of phenolic compounds in TEO and the antioxidant properties of mutton patties. Compared to TEO, TEOM more effectively inhibited fat and protein oxidation while enhancing color, sensory characteristics, and texture. Therefore, microencapsulation of TEO effectively enhances the quality and safety of mutton patties. 

The paper is well done but I have some remarks:

Moderate editing of English language required

- The conclusions have to expanded

Moderate editing of English language required

Round 2

Reviewer 2 Report

Dear Authors,

Thank you, the quality is much better now.

Best regards